# Galectins as Emerging Glyco-Checkpoints and Therapeutic Targets in Glioblastoma

**DOI:** 10.3390/ijms23010316

**Published:** 2021-12-28

**Authors:** Guillermo A. Videla-Richardson, Olivia Morris-Hanon, Nicolás I. Torres, Myrian I. Esquivel, Mariana B. Vera, Luisina B. Ripari, Diego O. Croci, Gustavo E. Sevlever, Gabriel A. Rabinovich

**Affiliations:** 1Laboratorio de Investigación Aplicada en Neurociencias (LIAN), Consejo Nacional de Investigaciones Científicas y Técnicas (CONICET), Fundación para la Lucha contra las Enfermedades Neurológicas de la Infancia (FLENI), Belén de Escobar B1625, Argentina; gvidela@fleni.org.ar (G.A.V.-R.); oliviamorrishanon@gmail.com (O.M.-H.); myriesq@hotmail.com (M.I.E.); mariana.belen.v.p@gmail.com (M.B.V.); luisina.ripari@gmail.com (L.B.R.); gsevlever@fleni.org.ar (G.E.S.); 2Laboratorio de Glicomedicina, Instituto de Biología y Medicina Experimental (IBYME), Consejo Nacional de Investigaciones Científicas y Técnicas (CONICET), Buenos Aires C1428, Argentina; nicotdroid@gmail.com; 3Laboratorio de Inmunopatología, Instituto de Histología y Embriología de Mendoza (IHEM), Consejo Nacional de Investigaciones Científicas y Técnicas (CONICET), Mendoza C5500, Argentina; dcroci@mendoza-conicet.gov.ar; 4Facultad de Ciencias Exactas y Naturales, Universidad de Buenos Aires, Buenos Aires C1428, Argentina

**Keywords:** central nervous system, glioblastoma, galectins, glycans, angiogenesis, immunomodulation, invasion, glyco-checkpoints, immunotherapy

## Abstract

Despite recent advances in diagnosis and treatment, glioblastoma (GBM) represents the most common and aggressive brain tumor in the adult population, urging identification of new rational therapeutic targets. Galectins, a family of glycan-binding proteins, are highly expressed in the tumor microenvironment (TME) and delineate prognosis and clinical outcome in patients with GBM. These endogenous lectins play key roles in different hallmarks of cancer by modulating tumor cell proliferation, oncogenic signaling, migration, vascularization and immunity. Additionally, they have emerged as mediators of resistance to different anticancer treatments, including chemotherapy, radiotherapy, immunotherapy, and antiangiogenic therapy. Particularly in GBM, galectins control tumor cell transformation and proliferation, reprogram tumor cell migration and invasion, promote vascularization, modulate cell death pathways, and shape the tumor-immune landscape by targeting myeloid, natural killer (NK), and CD8^+^ T cell compartments. Here, we discuss the role of galectins, particularly galectin-1, -3, -8, and -9, as emerging glyco-checkpoints that control different mechanisms associated with GBM progression, and discuss possible therapeutic opportunities based on inhibition of galectin-driven circuits, either alone or in combination with other treatment modalities.

## 1. An Introduction to Glioblastoma

CNS tumors continue to be a critical problem in oncology, with a 36% 5-year survival rate for malignant tumors [1]. The presence of a neoplastic mass in the CNS is always a clinical fact of certain aggressiveness in itself that is aggravated by the side effects of surgery, radiation, and chemotherapeutic agents [2]. The diffuse growth of glial tumors conspires against the possibility of an oncological resection, as it is usual in other organs [3]. Paradoxically, there has been an enormous advance in recent years in neuroimaging and molecular biology of CNS tumor development. This progress has not fully impacted the survival rate of malignant gliomas, which remains similar to several decades ago [4].

Gliomas, the most common brain and spinal cord tumors, originate from glial cells, including astrocytes, oligodendrocytes, ependymal cells, and microglia. Of these, astrocytomas are the most common types of gliomas in the pediatric, adolescent, and adult populations [5]. Traditional classifications have been made on a morphological basis, with a histogenetic criterion as the definition of the tumor’s cell of origin [6]. At present, this traditional concept is slowly being replaced by a classification based on molecular profiling of the different entities [7]. GBM, the most common and aggressive CNS malignancy with a 5-year overall survival of 5.6%, represents 47.7% of all primary malignant tumors of the CNS [1]. One of the constitutive features of these tumors is their heterogeneity and histopathological hallmarks involving nuclear atypia, necrosis, mitosis, and microvascular proliferation [8]. Primary GBM often lacks IDH1/2 mutations but exhibits epidermal growth factor receptor amplification, CDKN2A deletion, TERT mutations, and the combination of chromosome 7 gain with chromosome 10 loss. Interestingly, emerging evidence indicates that the morphological heterogeneity of these tumors could be explained by different molecular patterns. Due to the high mitotic index and genome instability, numerous subclones with different mutation patterns develop early. This dynamic heterogeneity probably makes it difficult to design a molecular-based therapy, highlighting the need to refine the search for molecular patterns to better reflect tumor evolution [7]. Nevertheless, the 2021 WHO classification [9] has adopted molecular identity profiles for most gliomas, summarizing the histological characteristics and developing a “layered system”, also known as Haarlem guidelines [10]. Thus, at present, the genetic definition of tumor lesions is a central component of tumor pathology. This change has been critical in the WHO classification since, for the first time in taxonomic conceptualization of brain tumor lesions, molecular biology is used as a central element in defining entities, especially glial and embryonic tumors [9]. Moreover, both the 2016 and 2021 WHO classifications have significantly impacted the search for actionable mutations to offer specific therapies in a field with scarce therapeutic options [9,11]. In this regard, a small subset of clinical trials has achieved therapeutic success in tumors with BRAF mutations and NTRK fusions, showing potential avenues for applying targeted therapies [12,13].

The recent identification of cancer stem cells (CSC) in GBM has revolutionized the field, as these cells have a profound influence in tumor progression, therapy resistance, and recurrence [14,15,16]. This concept is crucial in understanding the main biological issues that prevent therapeutic success in GBM [17]. Moreover, the infiltrative nature of glioma stem cells (GSC) appears to be a key factor that hinders success of localized therapies [18].

Another crucial diagnostic approach is the determination of DNA methylation patterns. Methylation is a dynamic epigenetic process that modulates the expression of the cellular transcriptome. Different methods have been described to evaluate the methylation status of gliomas, but the large-scale use and bioinformatics technologies have allowed the creation of a classifier that improves and refines diagnosis of this tumor type [19]. In the new WHO classification, seven new glioma types have been defined based on the specific methylation patterns [7].

Thus, advances in both molecular profiling and bioinformatics have generated a vast array of information on the biology of brain tumors in association with their clinical behavior [20]. This large set of data has generated a paradigm shift in the conceptualization and classification of CNS tumors [7]. Although this central model refers mainly to high-grade tumors, the impact of these discoveries applies broadly to all CNS tumors. The great challenge is to transfer this information to successful therapies that could impact patient’s survival and life quality. These include the search for novel therapeutic targets that could broadly influence different hallmarks of cancer, including tumor cell proliferation, migration, CSC survival, angiogenesis, and antitumor immunity.

## 2. Galectins in Tumor Progression: An Overview

Although glycans were originally conceived as biomarkers of tumor malignancy and disease progression, recent evidence highlights their functional roles as key regulators of primary tumor growth and metastasis, introducing the concept of tumor “glyco-checkpoints” in the TME [21,22]. Through either masking or unmasking specific glyco-epitopes for endogenous lectins, glycosylation can regulate a wide range of biological processes associated with tumor proliferation, survival, immunity, angiogenesis, and metastasis [23,24]. In this regard, the cellular glycome arises from the coordinated action of a set of glycan-modifying enzymes, including glycosyltransferases and glycosidases, which are spatiotemporally regulated at the genetic and epigenetic levels, depending on the cell type, activation and differentiation status [25,26].

Two major glycosylation pathways, namely *N*- and *O*-glycosylation, occur in almost every living organism, from bacteria to human cells [27,28] and other infectious agents [29]. The attachment of *N*-acetylglucosamine (GlcNAc) to an asparagine (Asn) residue within the consensus sequence -N-X-S/T, where X cannot be proline, represents an essential step in the *N*-glycosylation pathway. These glycans are composed of a GlcNac2 mannose (Man)3 core that can be then further modified to generate high-mannose, hybrid or complex *N*-glycans. Of particular interest, *N*-acetylglucosaminyltransferases (MGAT)-1, -2, -4, and -5 can generate *N*-acetylglucosaminyl branches, providing permissive sites for the elongation of *N*-acetyl-lactosamine (LacNAc; Galβ1-4GlcNAc) structures, which in turn represent the main ligands for galectins [30].

Compared to their normal counterparts, cancer cells and their associated microenvironment display altered glycosylation patterns, including increased frequency of complex branched *N*-glycans, truncated *O*-glycans, and changes in fucosylation or sialylation profiles [31]. Furthermore, the cellular glycome is substantially altered during immunological and vascular processes involving activation, proliferation, differentiation, and migration of immune and endothelial cells [32,33].

Lectins are dedicated glycan-binding proteins responsible for deciphering the biological information encrypted by cellular glycoconjugates [26,34,35]. Galectins, a family of soluble lectins, are released to the extracellular medium through a nonconventional ER-Golgi-independent pathway, unlike *C*-type lectin receptors (CLRs) and Siglecs, which are mostly cell-surface-associated glycan-binding proteins [36,37]. Fifteen members of the galectin family have been documented in diverse mammalian cells and tissues, where they play key roles in a wide range of biological processes [25]. Galectins have one or two highly conserved carbohydrate recognition domains (CRDs) that bind preferentially to the disaccharide LacNac present in *N*- and *O*-glycans, although subtle differences have been found in glycan-binding preferences of different members of the family [26]. Based on the CRD organization, mammalian galectins (and, by extension, galectins in vertebrates) have been classified into three types: “proto”, “chimera”, and “tandem-repeat”. Proto-type galectins, such as galectin-1, -2, -5, -7, -10, -11, -13, -14, and -15, contain one CRD per subunit and may behave as noncovalently linked homodimers. The chimera-type galectins (galectin-3) have a *C*-terminal CRD and an *N*-terminal domain rich in proline and glycine, which can mediate oligomerization (mostly trimers and pentamers). In tandem-repeat galectins such as galectin-4, -8, -9, and -12, two CRDs are joined by a functional linker peptide [38] (Figure 1). Individual members of the galectin family differ in their ability to interact with terminal or internal LacNAc structures and several modifications of LacNAc, including terminal sialic acid in the LacNAc sequence [30]; whereas addition of α2,3-linked sialic acid allows recognition by most galectins (particularly galectin-1, -8, and -9), and incorporation of α2,6-linked by the α2,6-sialyltransferase 1 (ST6GAL-1) prevents binding of galectin-1 to specific glycosylated receptors [25]. Additionally, numerous factors can alter the biological activity of galectins and their glycan-binding properties, including oxidative conditions, pH, and dimerization or oligomerization status [38]. In fact, galectin multivalency dictates their function and pleiotropic roles in almost all biological processes. Glycoproteins often bear multiple copies of the saccharide ligands that are recognized by multivalent galectins [38]. While galectin binding to a single saccharide ligand is typically a low-affinity interaction (association constants ~10^4^ M^−1^), the multivalent nature of galectin–saccharide interactions result in high overall avidity (association constants ~10^6^ M^−1^) [38]. The formation of multivalent galectin-glycan structures—often termed lattices—creates an extracellular scaffold that can cluster glycoproteins into homotypic or heterotypic aggregates that control receptor activation threshold, endocytosis, and signaling [39] (Figure 1). On the other hand, protein–protein interactions between galectins and several partners have been mostly described within the intracellular compartment [40].

Galectins are abundantly expressed in the TME, where they play essential roles by linking tumor, stromal, and immunological compartments [24,41]. They have been proposed to play key roles in virtually all hallmarks of cancer and metastasis [42,43]. This includes galectin-mediated programs that assist tumors in sustaining proliferation driven by oncogenic signaling circuits [44,45], evading growth suppressors [46,47], fostering inflammation [48,49,50], activating invasion processes such as epithelial–mesenchymal transition (EMT) and early dissemination [51,52,53], inducing angiogenesis [33,54,55,56], developing resistance to cell death [57,58], and avoiding immune destruction [59,60,61,62,63]. Particularly, galectins may control the immune landscape of several tumors by shaping effector and regulatory T-cell compartments [59,61,63,64,65], modulating myeloid cell functions, particularly those involving dendritic cells (DC), macrophages, and myeloid-derived suppressor cells (MDSC) [66,67,68,69], and influencing NK-cell-mediated cytotoxicity and cytokine production [70,71,72,73]. Moreover, recent evidence indicated a role for galectins as central mediators of resistance to anticancer therapies [74], including chemotherapy [75], antiangiogenic therapy [33], and immunotherapy [76,77]. In this article, we highlight the role of galectins in different processes linked to GBM progression, with a special focus on tumor proliferation, survival, invasion, vascularization, and immunity. Finally, we discuss the prognostic and therapeutic implications of galectin-driven circuits in brain malignancies, particularly GBM.

## 3. Galectins in CNS Malignancies: Regulated Expression and Prognostic Value

Galectins have been proposed to serve as reliable biomarkers of tumor progression in CNS malignancies, including GBM [78]. In early studies, galectin-1 expression correlated with the malignant potential of human GBM cells [79], and its prominent localization on the margins of these tumor cells contributed to delineate their invasive potential [80]. In this regard, stable inhibition of galectin-1 gene (*LGALS1*) expression altered the profile of several genes that either directly or indirectly influence adhesion, motility, and invasion of human GBM cells [81]. In this regard, this lectin is highly expressed in human gliomas, and plays key roles in the invasion of tumor astrocytes into the brain parenchyma [82].

More recently, by exploring different microenvironmental signatures in glioma databases, Chen et al. identified and validated the relevance of *LGALS1* as a major prognostic biomarker of tumor progression and immunosuppression [83]. Chou and colleagues validated the prognostic significance of this lectin in patients with GBM undergoing adjuvant radiotherapy [84], highlighting its clinical value as a possible biomarker of treatment responses in these tumor types.

On the other hand, pioneering studies have identified a strong correlation between galectin-3 expression and the malignant potential of CNS tumors [85]. Interestingly, this chimera-type lectin has been identified as an essential tool to distinguish distinct CNS tumors, including pilocytic astrocytomas from diffuse astrocytomas, and GBM from anaplastic oligodendrogliomas [86,87]. More recently, Takashima and colleagues proposed a gene signature composed of *LGALS3* (encoding galectin-3), the transcription factor GATA3, and CD276 in a prognosis assessment of GBM [88]. This effect was confirmed by Wang and colleagues, who verified the prognostic value of *LGALS3* in tumor risk and treatment responses in GBM [89].

To assess the predictive value of galectin-9 expression and its function, Yuan et al. performed comprehensive analysis of 1027 glioma patients using RNA-seq data and 986 patients with survival data. They found a strong upregulation of this lectin in GBM samples compared with normal brain tissues and lower-grade glioma. Patients with galectin-9 overexpression showed significantly shorter overall survival. Through cluster and correlation analysis, the authors found that expression of this lectin was highly associated with immune checkpoint pathways and M2 tumor-associated macrophages (TAM), highlighting the functional and relevance of this lectin as a potential biomarker of GBM progression [90]. Thus, a galectin signature, based on galectin-1, -3, and -9 expression levels may contribute to discriminate CNS tumors and delineate GBM progression, invasion, immunosuppression, and treatment responses. Further studies are warranted to integrate this signature with clinical, molecular, and histopathological determinants of disease evolution.

## 4. Galectins and Glioma Cell Proliferation

Unbridled proliferation is one of the defining hallmarks of cancer and the target of many chemotherapeutic treatments. Elucidating the proliferation pathways associated with glioma progression and therapy resistance has long been a central goal in neuro-oncology. Since their discovery, the role of galectins in proliferation has been widely recognized and described in a broad range of cell types, including breast epithelial cells [91], thyroid carcinoma cells [92], and neural stem cells [93]. The role of galectins in the regulation of tumor growth and proliferation largely depends on the target cell type. To illustrate this concept, while galectin-3 expression promotes the proliferation of lymphoma T cells [57] and breast cancer cells [94], it has growth-inhibitory activity in prostate cancer settings [95].

In the case of glioma, galectins mostly support proliferation of tumor cells. In this regard, Yamaoka and colleagues [79] studied the effect of silencing galectin-1 expression in the rat glioma cell line 9L, and found that the growth rate, in both monolayer and soft-agar colonies, decreased when galectin-1 levels were reduced. However, the authors did not assess whether growth rate attenuation was due to decreased proliferation, impaired survival, or both. On the other hand, Rorive et al. [82] evaluated the role of the exogenously added recombinant galectin-1 on cell kinetics of U87, U373, and H4 glioma cell lines, and found no significant effects either on mitosis, cell cycle progression, or growth rate, hinting at a possible intracellular mechanism that could underlie the effects of galectin-1 on glioma cell growth. Accordingly, while exogenous recombinant galectin-1 did not affect the growth rate of the A172 cell line, silencing of this lectin effectively impaired the proliferation of these cells [96]. Interestingly, galectin-1 knocked-down U87 cells showed increased expression of pro-proliferative factors such as proliferating cell nuclear antigen (PCNA) and cyclin D3, as well as antiproliferative factors such as p53 and p21, suggesting mixed effects of galectin-1 on glioma cell cycle progression [81]. In this context, Toussaint et al. [80] demonstrated no significant changes in tumor cell proliferation when they overexpressed galectin-1 in U87 cells. However, Guo et al. observed decreased proliferation in the galectin-1-silenced U251 cell line and increased proliferation in the galectin-1-overexpressing U87 cell line. They found that this effect was dependent on the FAM289, a newly discovered putative oncogene [97]. Although some of these results seem to be contradictory, it should be noted that many of these studies were conducted using different experimental approaches and distinct cellular models. In this regard, in an in vivo setting, silencing galectin-1 by an intranasal knockdown strategy resulted in reduced tumor Ki67 staining, indicating diminished proliferation rates than untreated tumors [75].

Interestingly, a number of studies highlighted a potential role of galectin-3 as an oncogenic factor that contributes to glioma cell growth. Ikemori et al. demonstrated that when nude mice were injected subcutaneously with U87MG cells transduced with a galectin-3 specific shRNA, tumors grew slower compared to mice injected with scrambled shRNA-transduced tumor cells. However, as the authors proposed, this could be an indirect effect not linked to proliferation, since in vitro silencing of this lectin did not affect the proliferation rate of glioma cells [98]. On the other hand, galectin-3 overexpression in the glioma T98G and U251 cell lines resulted in increased growth rates, suggesting that this lectin could promote glioma cell proliferation [89]. These results were in line with prior findings suggesting that galectin-3 expression in vivo correlates with cell proliferation measured by Ki67 in ethylnitrosourea (ENU)-induced tumors in Wister rats [99]. One possible mechanism underlying this effect might result from interaction of this lectin with Ras family members, an effect observed in the U251, U373, and U87 glioma cell lines [100].

Finally, Metz and colleagues reported a key role of galectin-8 in promoting glioma cell proliferation. They observed an increase in cell growth of the U87 cell line when recombinant galectin-8 was added to tumor cell cultures and verified changes in cell proliferation when this tandem-repeat lectin was downregulated in this cell line [101]. Thus, a coordinated network of galectins, particularly galectin-1, -3, and -8 might regulate tumor proliferation by acting in both the extracellular and intracellular compartments.

## 5. Impact of Galectins in GBM Migration and Invasion

Most gliomas from grade II to IV have diffuse borders and infiltrate surrounding tissue. Of these, GBM is the most extreme and aggressive primary brain tumor [11]. The invasive capacity of GBM cells is one of the most dangerous features of this tumor type and is ultimately the cause of death in most patients. Its diffuse nature makes its surgical resection a practical impossibility, allowing for the eventual tumor relapse occurring often adjacent to the resection margin, although also as distant satellite lesions [102,103]. In fact, GBM cells do not disseminate via hematic or lymphatic vessels; instead, they travel mostly along perivascular zones and white matter tracks such as neuroblasts, thus rarely metastasizing outside the brain [104,105]. GBM cells usually migrate as single cells or a chain of cells, and use distinctive structures called “microtubes” to aid in invasion [106,107]. GBM cells also share the classical hallmarks of invasive cells found in most cancers, such as their capacity to break off from the main tumor mass, heightened cell motility, and the ability to interact with and adhere to, but also degrade, the extracellular matrix to move across [103]. Invasion of GBM cells is then possible due to the processes of adhesion, motility, and ECM degradation, and the importance of galectins is clear, as they have been found to participate in all three processes.

As galectin-1 is highly expressed in GBM, its role in migration and invasion could be of great relevance in the outcome of this disease. Studies that analyzed the effects of galectin-1 on cell motility have consistently shown a promigratory effect displayed by this lectin. In 2001 Camby et al. [86] and Rorive et al. [82], using a video cell-tracking model for measuring cell movement, showed that galectin-1 was differentially expressed in the peritumoral region compared to the core, and that galectin-coated surfaces increased tumor cell migration. Later, Camby′s group [108] showed decreased cell motility when binding of extracellular galectin-1 was interrupted. These findings confirmed an extracellular role of galectin-1 in tumor cell migration. In this study, the authors silenced expression of galectin-1 using an antisense RNA and found that, as expected, migration was diminished. Consistent with these findings, treatment with recombinant galectin-1 [96] and overexpression of galectin-1 through an IRES vector [80] increased tumor cell migration. Toussaint et al. [80] also evaluated the effect of galectin-1 on invasion in vitro and in vivo and demonstrated that overexpressing galectin-1 in xenotransplanted U87MG cells changed the morphology of the tumors, showing increased invasion at the tumor–brain interface. The mechanisms behind this effect are still not completely understood; however, there is evidence that galectin-1 can affect the localization of β1 integrin to focal adhesion points, thus influencing migration and cell adhesion [109]. Guo et al. [97] also implicated *FAM289*, a newly discovered oncogene, as a mediator linking galectin-1 expression and its effects on cell migration and invasion. Although most of these results were observed using commercially available cell lines and not patient-derived primary tumors, there is little doubt of the role of galectin-1 in promoting GBM migration.

Unlike galectin-1, the role of galectin-3 in GBM migration and invasion is not so clear-cut. In noncancerous cells of the brain, the highest levels of this lectin are found in blood vessels, followed by white matter tissue and the grey matter [110]. This result is of interest, as GBM favors the perivascular and white matter tracks for migration. As with galectin-1, the expression levels of galectin-3 are higher in the more invasive sections of the tumor [86] and the cells associated with peripheral vessels [110]. In various glioma cell lines, variable results were seen in the adhesion of cells to galectin-3 substrate, with some cell lines showing minimal adhesion, while others displayed a significant amount. Of interest, the rat glioma cell line C6 did not adhere at all to galectin-3 [111]. In a cell motility assay, the three human cell lines tested (Hs683, T98G, and U373) showed a significant increase in their migratory capacity on galectin-3-coated substrates compared to plastic support [86]. These results indicated an extracellular role of galectin-3 on glioma migration. The effect of endogenous galectin-3 was studied by Debray et al. [112], who found interesting results using the U373 cell line. Silencing of galectin-3 in U373 cells decreased motility on uncoated plastic but accentuated this effect on laminin-coated plates. It also altered the cytoskeleton organization, increasing fibrillar actin and decreasing monomer actin. However, lower expression of galectin-3 did not affect invasion of this cell line through laminin or Matrigel. These conflicting results showed that the role of endogenous galectin-3 is not simple to elucidate. Moreover, since these results were observed only in the U373 cell line, it is unclear if they can extrapolate to other cell lines, which is especially important considering the heterogeneity seen in earlier studies. Of interest, in this same study, silencing galectin-3 was found to increase α6 and β1 integrins, both associated with cell adhesion.

Although expression levels of galectin-8 did not show differences between vessel walls, perivascular tumor astrocytes, and the tumor bulk in patient tumor samples, Camby et al. [86] described a slight increase in galectin-8 expression in invasive tumor fronts compared to the tumor core in xenografts generated from U87 and U373 cell lines. The authors also described an increase in migration of these cell lines on galectin-8-precoated plates compared to controls using a cell tracking method. These results were expanded upon by Metz et al. [101], who delved into the role of endogenous galectin-8 by silencing via shRNA and analyzing its exogenous function using its recombinant form as a soluble stimulant. In their results, they demonstrated that soluble galectin-8 increased migration of U87 cell lines in a transwell assay, although silencing galectin-8 did not affect the promigratory capacity of these cells, but clearly influenced tumor cell proliferation. This could be explained by the level of silencing achieved by shRNA (60%), as well as the fact that migration is less sensitive to lower expression levels of this lectin than other processes such as proliferation. Thus, galectin-8 appears to regulate GBM migration, at least in the cell lines studied.

Although galectin-9 is expressed at detectable levels in gliomas, to our knowledge, there is no published evidence on its role in GBM migration and invasion. Research into other tumor types showed a consistent pattern, as galectin-9 suppressed migration and invasion in various tumors studied such as hepatocellular carcinoma [113], melanoma, colon cancer [114], and gastric cancer [115]. Further studies should be conducted to further evaluate the mechanisms underlying modulation of GBM migration/invasion in response to galectin-9. Thus, galectins can modulate migration and invasion of GBM through mechanisms that still remain to be fully elucidated.

## 6. Galectins and Glioma Survival: Implications in Radio- and Chemoresistance

The standard of care protocol for newly diagnosed GBM is surgical resection followed by radio- and chemotherapy, in which temozolomide (TMZ) is the main chemotherapeutic drug [116]. However, because glioma cells are highly resistant to these treatments, tumor recurrence is still practically unavoidable. Therefore, to prevent GBM relapse, it would be critical to target biological pathways that support glioma cell viability, and thus sensitize tumor cells to radiotherapy and chemotherapeutic agents.

As previously mentioned, galectins can control different cell death programs extracellularly by interacting with specific glycosylated receptors or intracellularly by interfering with central signaling pathways [42]. Although galectins can promote or prevent cell death depending on the cellular localization or molecular context, they usually enhance survival of cancer cells, resulting in increased resistance to anticancer agents. For example, galectin-1 promotes chemoresistance of ovarian cancer [117], hepatocellular carcinoma [118], leukemia [119], and breast cancer [120]. Similarly, galectin-3 protects different tumor types from apoptosis triggered by different stimuli [58,121,122].

In this context, several studies have shown that galectin-1 downregulation favors glioma cell death and increases glioma sensitivity to chemotherapeutic drugs. Puchades et al. demonstrated that in vitro silencing of galectin-1 increased sensitivity of U87MG glioma cells to the topoisomerase inhibitor SN-38, and recombinant galectin-1 partially reversed this effect, suggesting that galectin-1 protects glioma cells via an extracellular mechanism [123]. In addition, ionizing radiation up-regulates galectin-1 expression in glioma cells, an effect that could limit their sensitivity to radiation therapy [96]. Moreover, Le Mercier et al. showed that TMZ increases galectin-1 expression in Hs683 glioma cells both in vitro and in vivo, and that galectin-1 downregulation enhances sensitivity to various cytotoxic drugs, including lomustine, carmustine, procarbazine, and vincristine. However, the authors found that galectin-1 silencing did not result in induction of apoptotic or autophagic pathways. Instead, knocking down galectin-1 decreased the expression of several genes associated with chemoresistance (*ORP150*, *HERP*, *GRP78/Bip*, *TRA*, *GADD45B*, *CYR61*, and *BNIP3L*), some of which are also involved in the unfolded protein response (UPR) to endoplasmic reticulum (ER) stress [124]. In vivo, Danhier et al. showed that U87MG orthotopic xenograft-bearing mice treated with TMZ plus anti-galectin-1 siRNA (delivered by chitosan lipid nanocapsules) exhibited an extended survival time compared with mice treated with TMZ alone [125]. Accordingly, Van Woensel and colleagues found that galectin-1 silencing via intranasal siRNA delivery extended mice survival and sensitized tumors to TMZ treatment in GL261 xenografts [75].

Recent evidence indicated that galectin-3 also controls glioma cell survival. Wang et al. reported that galectin-3 expression confers radio- and chemoresistance in the T98G and U251 GBM cell lines [89]. Furthermore, Ikemori et al. detected galectin-3 expression in pseudopalisades [98], which are regions that comprise cells that actively migrate away from hypoxic or necrotic foci [126], and found that hypoxia induced galectin-3 expression, which protects glioma cells from oxygen- and nutrient-deprivation-induced cell death. Accordingly, in vivo assays demonstrated that galectin-3 silenced-tumors, grafted in the flanks of immunodeficient mice, displayed decreased growth rates [98]. Similarly, Seguin et al. found that galectin-3 was present in pseudopalisates, and that hypoxia induced its expression. Importantly, they found that in the GBM mesenchymal subtype, galectin-3 was required for macropinocytosis, which in turn sustained GSC viability [127].

Finally, Metz et al. showed that galectin-8 protects U87 cells from apoptosis in vitro. Silencing galectin-8 with a specific shRNA resulted in a higher proportion of U87 cells in the sub-G1 phase and higher levels of active caspase-3 [101]. Thus, galectin-1, -3, and -8 are key modulators of GBM survival, controlling cellular sensitivity to chemotherapy and radiotherapy.

## 7. Galectins and Gliomas Angiogenesis

Excessive formation of disorganized blood vessels is one of the main characteristics of high-grade gliomas [128]. In brain tumors, the perivascular niche plays a fundamental role in tumor progression by secreting factors that maintain the properties of GSCs [129]. In fact, in high-grade gliomas, tumor cells actively migrate around the perivascular matrix, leading to tumor spread and complicating a definitive surgical removal of the tumor [11]. Strikingly, in three independent publications, it was described that GSCs were capable of directly transdifferentiating into endothelial cells [130,131,132].

Although different antiangiogenic strategies have been tested in preclinical and clinical trials, their efficacy in high-grade gliomas have not been fully encouraging. This limited success could be explained by different resistance mechanisms, including the expression of other proangiogenic factors, such as stromal-derived factor 1 (SDF1) and basic-fibroblast growth factor (bFGF) [133]. Alternatively, blocking VEGF-mediated proangiogenic signals often leads to an adaptive response called “angiogenic/invasive shift”, which allows tumor progression by inducing a more invasive phenotype [134]. Inhibition of angiogenesis led to hypoxia, promoting diverse cellular responses that culminated in a more invasive capacity of tumor cells [134,135,136]. This acquired resistance mechanisms urge the search of novel mediators that could simultaneously promote angiogenesis and invasion of GSCs.

Different studies have demonstrated the role of galectins in tumor vascularization [137,138,139,140]. Thijssen and colleagues have demonstrated a central role for galectin-1 in tumor angiogenesis [54,141]. Seeking possible mechanisms underlying this effect, our group found that galectin-1 was induced in response to hypoxia through hypoxia inducible factor-1 (HIF-1)-independent, nuclear factor kappa B (NF-kB)-dependent pathway [55] and promoted angiogenesis in several tumor types [55,142], particularly those resistant to vascular endothelial growth factor (VEGF)-targeted therapies [33]. At the molecular level, galectin-1 binds to complex *N*-glycans on vascular endothelial growth factor receptor 2 (VEGFR2) on endothelial cells, and triggers VEGF-like signaling through ERK1/2- and AKT-dependent pathways [33]. Interestingly, endothelial cells associated with tumors that are sensitive to anti-VEGF treatment are covered by α2,6-linked sialic acid, which prevents galectin-1 binding and subsequent angiogenesis. In contrast, endothelial cells associated with tumors that are resistant to anti-VEGF are highly enriched in complex branched *N*-glycans and display low levels of α2,6 sialylation, which favors Gal-1 binding, VEGFR2 signaling, and compensatory angiogenesis [33]. Thus, targeting the Gal-1-glycan axis emerges as a potential mechanism to counteract aberrant angiogenesis in anti-VEGF refractory tumors.

In gliomas, galectin-1 also contributes to tumor angiogenesis. Silencing galectin-1 expression in the Hs683 glioma cell line impaired endothelial cell migration and tubulogenesis in vitro. Moreover, in vivo silencing of galectin-1 in Hs683 orthotopic xenografts significantly decreased vessel density of tumors, suggesting that galectin-1 is a central player in glioma angiogenesis. Silencing galectin-1 expression lowered expression of hypoxia-responsive genes involved in angiogenesis and significantly reduced VEGF secretion [143]. The authors further showed that galectin-1 silencing resulted in downregulation of BEX2, a gene that is highly expressed in the embryonic brain. In turn, reduction in BEX2 expression resulted in less-vascularized tumors and increased survival of Hs683 orthotopic xenograft-bearing mice. Interestingly, decreasing BEX2 expression also impaired the migratory capacity of Hs683 cells, without affecting cell growth and survival [144]. Hence, modulation BEX2 activity could explain, at least in part, both the proangiogenic and promigratory roles of galectin-1 during glioma progression. In this regard, another study suggested that galectin-3 was also induced under hypoxic conditions through HIF-1α- and NF-kB-dependent mechanisms and could protect glioma cells from cell death induced by hypoxic and nutrient deprivation conditions [98]. Given the involvement of galectin-3 in tumor angiogenesis [56], these results suggested a central role for this lectin during GBM progression.

## 8. Impact of Galectins in Anti-GBM Immunity

GBM, the most immunosuppressive CNS cancer, is generally regarded as a “cold tumor”, characterized by low number of tumor-infiltrating lymphocytes (TILs), low frequency of tumor-associated antigens, poor inflammatory capacity, and increased accumulation of myeloid populations with a predominant protumorigenic profile. This pronounced immunosuppressive phenotype is responsible for hindering the clinical success of most immunotherapeutic modalities in GBM patients [145,146].

Compelling evidence supports the immunosuppressive role of galectin-1 in different tumor types and its ability to promote tumor-immune escape. This lectin contributes to immune evasion by regulating the fate and function of different lymphoid and myeloid cells [147]. As mentioned above, galectin-1 fosters immunosuppressive programs by selectively controlling the fate of Th1, Th17, and CD8^+^ T cells both in peripheral and mucosal compartments [64], inhibiting transendothelial T-cell migration [148], and promoting the differentiation of CD4^+^ and CD8^+^ Tregs [63,149]. Furthermore, this lectin can reprogram the tolerogenic activity of DCs [66] and polarize macrophages toward a protumoral/proresolving M2 profile [150,151]. Additionally, within the CNS, galectin-1 can deactivate classically activated M1-type microglia by interacting with core-2 *O*-glycan structures on CD45 and modulating its phosphatase activity [152].

Interestingly, a number of studies have documented the central role of galectin-1 in suppression of antitumor immune responses during GBM progression. In a series of publications, Lowenstein and colleagues explored the role of galectin-1 in antitumor innate responses in GBM [71,73,153]. These foundational observations were based on inoculation of galectin-1 knocked-down glioma cells in both immunocompetent and immunocompromised mice. The authors found an early increase (48 h postengraftment) of an immature myeloid population capable of influencing NK cell activity required for tumor killing. This response allowed the complete eradication of the tumor in less than 14 days, even before the onset of an adaptive immune response. Notably, increased frequency of myeloid-derived inflammatory cells was associated with higher expression of the CCL2 chemokine, as well as multiple cellular-stress-related molecules [71,153]. To further understand the molecular mechanisms underlying this myeloid-NK cell cross-talk, this group recently demonstrated that, in the absence of galectin-1, malignant cells release exosomes containing miR-1983. This t-RNA-derived miRNA acts as an endogenous TLR7 ligand in classical and plasmacytoid DCs, which, by releasing IFN-β, activate NK cells. Subsequently, NK cells release granzyme B (GmzB) and perforins, inducing apoptosis of malignant cells. However, in tumor cells that overexpress galectin-1, release of miR-1983-containing exosomes is strongly diminished, and this process is subsequently repressed [73]. Collectively, these results highlighted the importance of innate immune mechanisms in the elimination of GBM and the essential role of galectin-1 in limiting these early antitumor responses

On the other hand, De Vleeschouwer and colleagues studied the role of galectin-1 at later stages of antitumor immune response. Using a lentiviral approach to downregulate the expression of galectin-1 in GL261 glioma cells, the authors found that, 14 days after tumor inoculation, the proportion of TAMs and MDSCs significantly decreased in the TME, and this was associated with lower expression of CCL2 in tumor cells. Whereas no significant differences were observed in the number of TILs nor in the expression of costimulatory or coinhibitory molecules, or in the percentage of Tregs; mice bearing galectin-1-silenced tumors showed higher survival rates, and this effect was partially reversed when CD8^+^ T cells were depleted in vivo, thus underscoring the importance of cytotoxic effector T cells in GBM antitumor immunity. These protumorigenic effects relied on tumor-derived but not host-derived galectin-1 [67]. Collectively, these studies demonstrated that tumor-derived galectin-1 impacts both innate and adaptive arms of antitumor immune responses. To examine the therapeutic relevance of these findings, the authors prepared chitosan nanoparticles loaded with a galectin-1-specific siRNA which were administered intranasally to mice transplanted with GL261 glioma cells. Using this approach, galectin-1 was silenced not only in tumor cells but also in the TME. At 20 days after tumor inoculation, a considerable polarization of TAMs from an M2 to an M1 phenotype, as well as a reduction of monocytic MDSCs was evidenced. Additionally, in this setting, the authors detected an increased number of CD4^+^ and CD8^+^ TILs and a reduced frequency of Foxp3^+^ Tregs [75]. Thus, galectin-1 shapes the glioma immune landscape by targeting both innate and adaptive immune compartments. Remarkably, this study further highlighted a role for galectin-1 as a mechanism of resistance to immunotherapeutic strategies. When DC vaccines were applied to mice bearing GL261 tumors, a median survival benefit from 20.5 days (without vaccines) to 42 days (with vaccines) was observed. Moreover, when DC vaccines were combined with chitosan nanoparticles loaded with anti-Gal1 siRNA, survival rates were prolonged to 53 days [75]. This combinatory strategy also demonstrated the development of immunological memory against glioma cells [67]. Increased survival was also evidenced by combining anti-galectin-1 siRNA with temozolomide (from 32 to 53 days) or anti-PD-1 monoclonal antibodies (from 30 to 51.5 days) [75]. Thus, targeting galectin-1, either alone or in combination with immunotherapeutic or chemotherapeutic agents, improves antitumor immune responses by reprogramming both lymphoid and myeloid cells.

Galectin-9, a tandem-repeat galectin, can either promote or suppress tumor growth and metastasis, depending on whether it acts on immune or tumor cells. Acting on different immune cells, galectin-9 contributes to immune escape. In fact, when it binds to Th1 lymphocytes, galectin-9 functions as a ligand for T-cell immunoglobulin and mucin domain-containing 3 (TIM-3), triggering apoptosis and negatively modulating Th1 immunity [154]. Additionally, galectin-9 induces apoptosis of cytotoxic CD8^+^ T cells [155] and inhibits NK cell-mediated cytotoxicity [156] via TIM-3-dependent mechanisms. In pancreatic adenocarcinoma, galectin-9 promotes an immunosuppressive phenotype in TAMs through binding to the *C*-type lectin Dectin-1 [157]. In addition, through association with CD44, galectin-9 promotes Treg differentiation and accentuates their immunosuppressive function [158]. Conversely, when galectin-9 acts directly on tumor cells, it may inhibit tumor growth. This is the case for multiple myeloma cells, in which galectin-9 acts as a potent antiproliferative signal [159]. Moreover, in melanoma and chronic myelogenous leukemia, this lectin promotes tumor cell apoptosis [160,161]. On the other hand, in mouse models of melanoma and colon cancer, galectin-9 suppresses tumor metastasis by inhibiting tumor cell adhesion to extracellular matrices and endothelium [114].

In gliomas, galectin-9 inhibits antitumor immune responses, thus favoring tumor progression. Wang et al. studied the impact of galectin-9 on DCs of GBM patients. The authors analyzed tumor-derived exosomes and immune cells in the cerebrospinal fluid (CSF) of patients. They found that DCs from GBM patients exhibited lower expression of molecules involved in antigen presentation and DC maturation. Importantly, they found that exosome-derived galectin-9 could induce TIM-3 expression on DCs present in the CSF and reprogram these cells toward an immature tolerogenic phenotype. Accordingly, mice transplanted with Lgals9-deficient glioma cells showed improved survival, increased activation of mature DCs, and enhanced CD8^+^ T-cell proliferation and function [162]. In order to study the role of the galectin-9/TIM-3 pathway on T lymphocytes, Liu et al. analyzed the association between the expression levels of these molecules and the clinicopathological features of glioma patients. They observed overexpression of TIM-3, both in peripheral blood T cells and TILs, and increased frequency of TIM3^+^ TILs positively correlated with tumor grade in glioma patients. Similarly, although weak expression of galectin-9 was detected in healthy brains, there was a significant increase in its expression in tumor biopsies, especially in grade IV gliomas. Finally, the authors reported a correlation between galectin-9 expression in tumor cells and TIM-3 expression on TILs [163]. In line with these findings, Yuan et al. also found increased galectin-9 expression in glioma, which correlated with shorter overall survival and higher tumor grade. Through metagene analysis, the authors found that galectin-9 levels correlated with the expression of multiple immune checkpoint molecules and M2-like genes [90]. By studying human diffuse astrocytomas samples (grades III and IV), Sorensen et al. found lower expression of galectin-9 and TIM-3 in IDH mutant compared to IDH wild-type gliomas. Using multiplex chromogenic immunohistochemistry and stereological-based cell counting, they found that the majority of the TIM-3^+^ or galectin-9^+^ cells corresponded to TAMs rather than tumor cells, and that the sparse TIM-3^+^ TILs preferentially interacted with galectin-9^+^ TAMs. Finally, in silico analysis revealed that tumors that exhibited higher expression of TIM-3 also displayed higher expression levels of genes related to leukocyte migration, macrophage polarization, and cytokine secretion [164]. As a whole, these studies suggested that the galectin-9/TIM-3 axis acts as an immunosuppressive modulator of the innate and adaptive immune responses. Therefore, although the intrinsic role of galectin-9 on glioma cells has not yet been elucidated, its effects on the glioma microenvironment suggest a potential role of this lectin as an immunotherapeutic target, alternative to the PD1/PD-L1 pathway.

Galectin-3, a chimera-type lectin, displays both pro- and anti-inflammatory effects, either promoting or limiting immune responses [40]. The versatility and plasticity of this lectin is explained not only by its ability to crosslink and modulate a wide variety of surface receptors, but also by its capacity to exert different intracellular functions [165]. Indeed, the cellular location of galectin-3 defines whether it will function as an antiapoptotic or proapoptotic factor [57,166]. Accordingly, galectin-3 has several roles in the TME, displaying both proinflammatory and immunosuppressive functions. As a proinflammatory molecule, extracellular galectin-3 favors neutrophil activation [167] and acts as a chemoattractant of human monocytes and macrophages [168]. In addition, it promotes superoxide production [169] and augments microglia activation [170]. However, galectin-3 may also function as a negative regulator of immune responses. While it can reprogram macrophages toward an M2-like phenotype [171], this lectin suppresses CD8^+^ T-cell function and decreases the number of circulating plasmacytoid DCs through binding to lymphocyte activation gene-3 (LAG-3) [172].

To date, the role of galectin-3 in the glioma TME has been scarcely studied. Chen et al. [173] identified a new galectin-3 binding glycoprotein, CHI3L1, which was considerably upregulated in GBM. CHI3L1 is secreted by tumor cells and binds to galectin-3 in the TME. The resulting CHI3L1/galectin-3 complex enhances migration and infiltration of macrophages and their polarization toward an M2-like protumoral phenotype, finally leading to T-cell dysfunction. In addition, the galectin-3-binding protein (Gal3BP) competes with galectin-3 in binding to CHI3L1, thus inhibiting CHI3L1 modulation by TAMs. Interestingly, the authors developed a Gal3BP Mimetic Peptide (GMP) that interrupts the CHI3L1/galectin-3 interaction. Notably, in vivo administration of this peptide reduced tumor size, increased overall mice survival, reduced the number of M2-like TAMs, and increased the frequency of cytotoxic T cells. GMP also upregulated PD-1 and CTLA-4 expression on CD8^+^ T cells, suggesting that CHI3L1/galectin-3 disruption could improve the efficacy of immune checkpoint inhibitors in GBM. Thus, galectins-1, -3, and -9 emerged as multifunctional glyco-checkpoints in GBM by shaping both lymphoid and myeloid compartments.

## 9. Conclusions and Future Perspectives

In this review, we discussed the central role of galectins, particularly galectin-1, -3, -8, and -9 in different hallmarks of GBM, their function as emerging glyco-checkpoints in CNS malignancies, and their therapeutic potential in this devastating disease. Remarkably, galectins can regulate glioma proliferation, survival, migration, angiogenesis, and tumor immunity (Figure 2). However, in spite of considerable progress, several questions remain to be addressed regarding the glycan-dependency of these functions, the compensatory role of individual members of the family, the relevance of extracellular versus intracellular activities of these proteins, and their prognostic value as possible biomarkers of GBM progression and invasiveness. Given the broad protumorigenic activity of galectin-driven circuits, different strategies have been proposed to neutralize galectins in the TME, including glycan-based small molecule inhibitors, large polysaccharides, antagonistic peptides, aptamers, and anti-galectin monoclonal antibodies [174,175,176,177]. Interestingly, some of these inhibitors are currently under evaluation in phase I or II clinical trials, including TD139, a galectin-3 inhibitor in patients with idiopathic pulmonary fibrosis; GR-MD-02, a pectin-derived galactoarabino-rhamnogalacturonan polysaccharide in patients with advanced melanoma; and OTX008, a caliraxene-based galectin inhibitor in patients with advanced solid tumors [178]. Future strategies should be aimed at refining the specificity of galectin-directed agents to avoid “off-target” and side effects, while maintaining a strong affinity and broad therapeutic potency to be further evaluated in preclinical models and clinical trials. This should include strategies to ensure full penetration across the blood–brain barrier, including the generation of anti-galectin single-chain fragment variable (scFv) monoclonal antibodies and small molecule glycans, peptides, and peptidomimetics. Additionally, in vivo studies using galectin-targeted agents should be performed in combination with antiangiogenic therapies, immunotherapy, radiotherapy, or chemotherapy to fully embrace their therapeutic potential. Finally, given the broad protumorigenic functions of galectins, particularly galectin-1, within the CNS, new experiments should be conducted to evaluate the broad consequences of their inhibition in the proliferation, survival, migration, angiogenesis, and antitumor capacity of GBM.

## Figures and Tables

**Figure 1 ijms-23-00316-f001:**
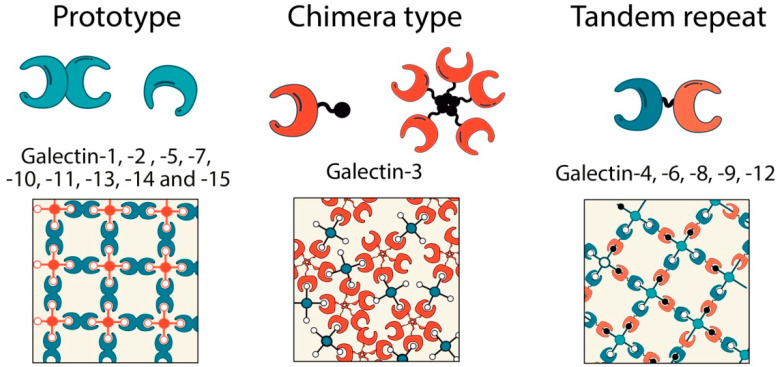
Schematic representation and structural classification of galectins. Galectin family members are subdivided into three groups proto-type galectins containing one carbohydrate-recognition domain (CRD) (including Gal1, Gal2, Gal3, Gal5, Gal7, Gal10, Gal11, Gal13, Gal14, and Gal15); galectin-3, a unique chimaera-type galectin consisting of repeats of proline- and glycine-rich short stretches fused onto the CRD; and tandem-repeat-type galectins (Gal4, Gal6, Gal8, Gal9, and Gal12), which contain two distinct CRDs connected by a linker of up to 70 amino acids. Lower panels represent hypothetical lattice formations of each group. Galectin’s bindings can be monovalent, bivalent, or multivalent regarding their carbohydrate-binding activities: Proto-type one-CRD galectins often exist as dimers; galectin-3 undergoes conformational changes after binding multivalent ligands that enable its oligomerization as pentamers, and two-CRD tandem-repeat galectins have two carbohydrate-binding sites that form lattices with multivalent glycoconjugates.

**Figure 2 ijms-23-00316-f002:**
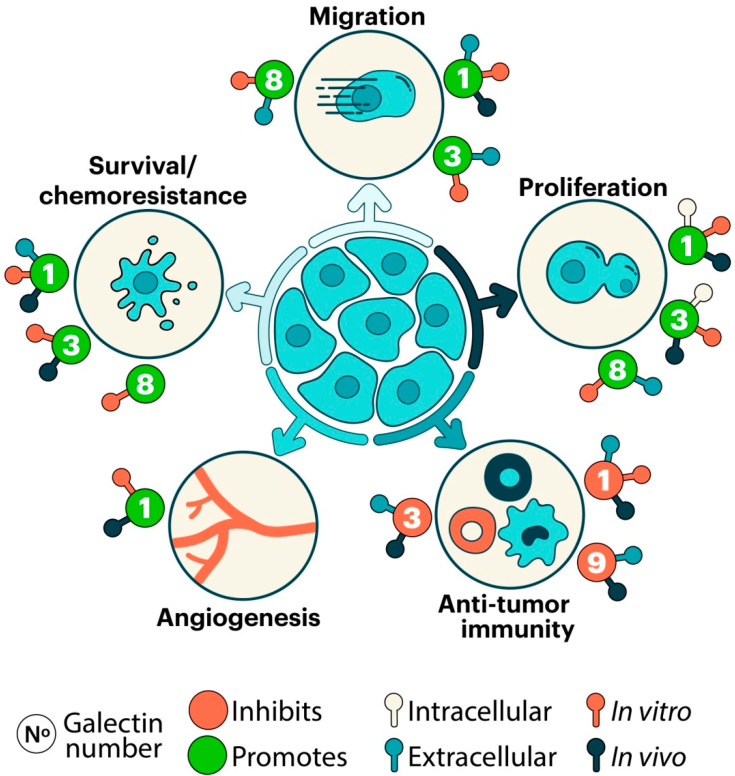
Galectins in glioma progression and tumor-associated inflammatory responses. Representation of biological functions of galectins in glioma cells (in vitro), or in animal models (in vivo). Galectins can promote a wide range of biological functions extracellularly by clustering multiple multivalent glycoconjugates, or intracellularly by modulating several signal transduction pathways, triggering intracellular oncogenic (proliferative and/or antiapoptotic) signaling. They can also bridge two cells of the same or different types, and bridge cells to extracellular matrix proteins, promoting migration and metastasis. In addition, galectin binding to extracellular glycoconjugates can induce angiogenesis, and affects immune responses in tumor microenvironments.

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
