# Peer review of "Galectins as Emerging Glyco-Checkpoints and Therapeutic Targets in Glioblastoma"

_ijms, 2021, doi:10.3390/ijms23010316_

Round 1

Reviewer 1 Report

Dear authors,

in lines 56 and 57 is stated, "GBM lacks IDH 1/2 mutations ...". IDH mutation state is important for molecular classification of DBM and is also a prognostic factor. So this must be clarified.

Author Response

Answers to Reviewer #1

  1. In lines 56 and 57 is stated, "GBM lacks IDH 1/2 mutations ...". IDH mutation state is important for molecular classification of DBM and is also a prognostic factor. So this must be clarified.

Response: We thank you very much for this comment. You will find the corresponding correction in the current version of the manuscript. In this sentence, we wished to mention the most frequent mutations observed in glioblastoma (GBM). For this reason, we refer only to primary GBM, which represents approximately 90% of grade IV gliomas. Therefore, to correct our mistake we have now incorporated the word "primary". Also, we noted that it would increase the clarity of the manuscript if the order of this sentence and the previous one is reversed. This change is also present in the new version of our manuscript.

Reviewer 2 Report

This review discusses the role of galactins in various critical biological processes central to glioblastoma progression. It is very concise, well written, and covers almost every aspect of galactins in this glioblastoma. I thoroughly enjoyed reading this review. However, I have a few minor comments, and a suggestion that I feel will add value to this article: 

Page 3, L112: Authors talk about N-glycosylation and glycosyltransferases. However, they are citing another review here. I would suggest citing original papers here.

Page 3 L146: Here, the author talks about galectin's multivalency and how it dictates their function and pleiotropic roles in almost all biological processes. Is there any repost of binding diversity and/or binding co-operativity in galectins? In other words, I am just curious if oligomerization is known to increase/decrease binding affinity for specific glycans? 

I feel adding a paragraph about the current state-of-the-art of galectin inhibitors and the challenges involved in designing glycomimetics against the galectins will be valuable to review. 

Author Response

Answers to Reviewer 2

This review discusses the role of galactins in various critical biological processes central to glioblastoma progression. It is very concise, well written, and covers almost every aspect of galactins in this glioblastoma. I thoroughly enjoyed reading this review. However, I have a few minor comments, and a suggestion that I feel will add value to this article:

We thank you very much the reviewer for encouraging comments

  1. Page 3, L112: Authors talk about N-glycosylation and glycosyltransferases. However, they are citing another review here. I would suggest citing original papers here.

Answer: Following the suggestion of the reviewer , two review articles on glycans and glycosyltrasnferases have been replaced for the following original articles:

  1. Lagana A, Goetz JG, Cheung P, Raz A, Dennis JW, Nabi IR. (2006) Galectin binding to Mgat5-modified N-glycans regulates fibronectin matrix remodeling in tumor cells. Mol Cell Biol. 26(8):3181-3193. doi: 10.1128/MCB.26.8.3181-3193.2006.

  1. Hernandez JD, Klein J, Van Dyken SJ, Marth JD, Baum LG. (2007) T-cell activation results in microheterogeneous changes in glycosylation of CD45. Int Immunol. 19(7):847-856. doi: 10.1093/intimm/dxm053.

  1. Page 3 L146: Here, the author talks about galectin's multivalency and how it dictates their function and pleiotropic roles in almost all biological processes. Is there any repost of binding diversity and/or binding co-operativity in galectins? In other words, I am just curious if oligomerization is known to increase/decrease binding affinity for specific glycans?

To clarify the importance of galectin multivalency in modulating affinity to specific glycans, and describe this issue appropriately, we have completed the following sentence:

"… In fact, galectin multivalency dictates their function and pleiotropic roles in almost all biological processes. Glycoproteins often bear multiple copies of the saccharide ligands that are recognized by multivalent galectins [38]. While galectin binding to a single saccharide ligand is typically a low-affinity interaction (association constants ~104 M-1), the multivalent nature of galectin–saccharide interactions results in high overall avidity (association constants ~106 M-1) [38]. The formation of multivalent galectin-glycan structures -often termed lattices- creates an extracellular scaffold that can cluster glycoproteins into homotypic or heterotypic aggregates that control receptor activation threshold, endocytosis, and signaling [39] (Figure 1)…"

  1. I feel adding a paragraph about the current state-of-the-art of galectin inhibitors and the challenges involved in designing glycomimetics against the galectins will be valuable to review.

Following the suggestion of the reviewer, we have completed the following paragraph:

"Given the broad pro-tumorigenic activity of galectin-driven circuits, different strategies have been proposed to neutralize galectins in the TME, including glycan-based small molecule inhibitors, large polysaccharides, antagonistic peptides, aptamers and anti-galectin monoclonal antibodies [174-177]. Interestingly, some of these inhibitors are currently under evaluation in phase I or II clinical trials including TD139, a galectin-3 inhibitor in patients with idiopathic pulmonary fibrosis, GR-MD-02, a pectin-derived galactoarabino-rhamnogalacturonan polysaccharide in patients with advanced melanoma and OTX008, a caliraxene-based galectin inhibitor in patients with advanced solid tumors [178]. Future strategies should be aimed at refining the specificity of galectin-directed agents to avoid 'off-target' and side effects, while keeping a strong affinity and broad therapeutic potency to be further evaluated in pre-clinical models and clinical trials."

Accordingly, the following reference was incorporated at the end of the article:

  1. Girard, A. and Magnani, J. L. (2018) Clinical trials and applications of Galectin antagonists. Trends Glycosci. Glycotechnol. 30, SE211–SE220..
